# Multi-Species Probiotic Strain Mixture Enhances Intestinal Barrier Function by Regulating Inflammation and Tight Junctions in Lipopolysaccharides Stimulated Caco-2 Cells

**DOI:** 10.3390/microorganisms11030656

**Published:** 2023-03-03

**Authors:** Hyeontak Han, Yeji You, Soyoung Cha, Tae-Rahk Kim, Minn Sohn, Jeseong Park

**Affiliations:** Center for Research and Development, Lactomason Co., Ltd., Jinju 52840, Republic of Korea

**Keywords:** probiotics, leaky gut syndrome, intestinal barrier function, tight junctions, anti-inflammation, multi-species probiotic mixture

## Abstract

Although leaky gut syndrome is not recognized as an official diagnosis for human diseases, it is now believed that dysfunction of the cell barrier causes increased permeability of intestinal epithelial cells leading to this condition. Probiotics have been widely used to improve gut health, and studies have investigated the relevance of protecting the intestinal barrier by taking probiotic strains in vitro and in vivo. However, most studies have restricted the use of single or several probiotic strains and do not consider commercially available probiotic products composed of multi-species. In this study, we provide experimental evidence that a multi-species probiotic mixture composed of eight different strains and a heat-treated probiotic strain is effective in preventing leaky gut conditions. We employed an in vitro co-culture model system utilizing two different differentiated cell lines to mimic human intestinal tissue. The integrity of epithelial barrier function was protected by the preserving the occludin protein level and activating the AMPK signaling pathway, associated with tight junctions (TJs), through treatment with the probiotic strain mixture in Caco-2 cells. Moreover, we confirmed that application of the multi-species probiotic mixture reduced the expression of proinflammatory cytokine genes by inhibiting NFκB signaling pathway when artificial inflammation was induced in an in vitro co-culture model system. Finally, we proved that the epithelial permeability measured by trans-epithelial electrical resistance (TEER) was significantly decreased in the probiotic mixture treated cells, indicating that the integrity of the epithelial barrier function was not compromised. The multi-species probiotic strain mixture exhibited the protective effect on the integrity of intestinal barrier function via enhancing TJ complexes and reducing inflammatory responses in the human intestinal cells.

## 1. Introduction

The human gastrointestinal tract is responsible for absorbing nutrients from ingested foods and providing energy to the body [1]. The intestinal barrier, consisting of a single layer of cells, is the first line of defense against harmful substances and microorganisms originating from foods. Maintaining the selective permeability of intestinal barrier to useful nutrients while preventing toxins and harmful microbes from entering the circulatory system is critical for overall health [2]. Leaky gut, a recently proposed hypothetical condition with increased permeability of the intestinal barrier, is based on the theory that the trespass of microbes through a weakened epithelial barrier may cause various illnesses. Interestingly, diseases triggered by leaky gut do not seem to be restricted to inflammatory bowel disease (IBD), celiac disease, and colon cancer in the gastrointestinal tract [3,4]. Piling evidence suggests that it affects mental mood, such as depression, suicidal thoughts, chronic fatigue, schizophrenia, and autistic spectrum [5,6,7].

The intestinal epithelium is constantly stimulated by various bioactive molecules such as food and microbe-originated antigens, metabolites, and even psychological stress. Therefore, proper control of immune responses is key for maintaining normal intestinal function.

Gut microbiota has been studied for its relevance to intestinal health, since the intestine is known to have the largest number of microbes among other human organs. One of the well-known functions of intestinal microbes is to aid ingested food digestion by processing plant fibers into absorbable short-chain fatty acids (SCFAs) and vitamins [8]. However, accumulating evidence has indicated that microbes inside the human gut play specialized roles in maintaining the integrity of the physical intestinal barrier, and the mode of achieving this is believed to be the generation of microbial metabolites and bacterial components [9]. For instance, butyrate, a SCFA resulting from bacterial fermentation, is known to be beneficial for potentiating TJs, which are crucial multiprotein complexes that regulate the permeability of the intestinal barrier at physiological concentrations [10,11]. Moreover, bacterial components such as lipopeptides enhance TJ function by stabilizing ZO-1 protein levels by activating protein kinase C as lipopeptides that bind Toll-like receptors (TLRs) on intestinal epithelial cells [12].

For this reason, several studies have assessed whether probiotic strains enhance the intestinal barrier function. In addition to using probiotic metabolites, Anderson et al. utilized the living strain itself to study TJ function in a Caco-2 cell line, and the study showed higher trans-epithelial electrical resistance (TEER), which is a measurement of integrity of barrier function, with increased TJ-related protein levels by using *Lactiplantibacillus plantarum* (previously *Lactobacillus plantarum*) 452 strain [13]. Similar approaches using other single or mixed probiotic strains have generated promising results, and studies have been carried out in vitro [14,15,16,17,18] and in vivo including clinical trials [19,20,21,22,23].

Balance WithinTM Immunity (hereinafter referred to as “BWI mix”) is a commercial probiotic supplement from Amway Korea Ltd. It contains eight live probiotic strains, six of which are patented for their anti-inflammatory properties. *Limosilactobacillus reuteri* LM1071 and *Streptococcus thermophilus* LM1012 have both been reported to have anti-inflammatory effects [24,25,26]. Additionally, the immune-boosting effect of a heat-killed material from *Lactiplantibacillus plantarum* LM1004 has been proven through in vitro and in vivo results, including a human clinical trial [27,28,29]. Therefore, the primary goal of the formula was to maintain homeostasis of human immunity with anti-inflammatory and immune-boosting strains.

The purpose of our study was to evaluate the impact of BWI mix, a multi-species probiotic mixture, on intestinal barrier function. To investigate this effect, we utilized an in vitro cell line model that mimics the physical environment of the gut. By co-culturing both differentiated Caco-2 cells and THP-1 cells, we measured TEER to assess the integrity of the cellular barrier, and we monitored the expression level of occludin to evaluate the formation of TJ complexes. Finally, we analyzed the cellular signaling pathways involved.

## 2. Materials and Methods

### 2.1. Preparation of Probiotic Mixture

The BWI formula is composed of 7 live probiotic species [*Lactiplantibacillus plantarum* LM1001 (KCCM 42959) (47.82%), *Limosilactobacillus reuteri* LM1071 (KCCM12650P) (19.79%), *Bifidobacterium animalis ssp. lactis* HEM 20-01 (KCTC 14143BP) (6.59%), *Bifidobacterium animalis ssp. lactis* LM1017 (KCCM12629P) (6.60%), *Lactococcus lactis* LM1009 (KCCM 80146) (8.24%), *Bifidobacterium longum* LM1024 (KCCM 80145) (0.25%), *Limosilactobacillus fermentum* HEM 1036 (KCTC 13978BP) (1.65%), and *Streptococcus thermophilus* LM1012 (KFCC 11771P) (0.82%), (total 8 live strains)] and a heat-killed material from *Lactiplantibacillus plantarum* LM1004 (KCCM 43246) (8.24%). The same ratio of original BWI formular was applied to prepare the multi-species probiotic mixture (BWI mix) through this study.

### 2.2. Caco-2 Cell Culture

Caco-2 cells (ATCC, Manassas, Virginia, No. HTB-37), a colorectal adenocarcinoma cell line, were cultured in Dulbecco’s modified Eagle’s medium (Sigma-Aldrich, St. Louis, MO, USA, cat. no. D5796) containing 10% fetal bovine serum (Sigma-Aldrich, St. Louis, MO, USA, cat. no. F2442) and penicillin and streptomycin (Thermo Scientific, Waltham, MA, USA, cat. no. 15140122). The cells were maintained in a humidified atmosphere containing 5% CO_2_ at 37 °C. The culture medium was changed every 2–3 days, and cells were passaged when confluence reached about 70–80%.

### 2.3. THP-1 Cell Culture

Human THP-1 monocytic cells (ATCC, Manassas, VA, USA, No. TIB-202) were cultured in T-75 flasks containing RPMI 1640 (Sigma-Aldrich, St. Louis, MO, USA, cat. no. R8758), 10% FBS, and 1% penicillin/streptomycin. THP-1 cells were incubated at in a humidified atmosphere containing 5% CO_2_ at 37 °C. Cells were subcultured when they reached a concentration of 1 × 10^6^ cells/mL.

### 2.4. THP-1 Differentiation

THP-1 cells were differentiated in culture medium containing 150 nM phorbol 12-myristate-13-acetate (PMA; Sigma-Aldrich, St. Louis, MO, USA, cat. no. P8139) for 24 h, followed by 24 h of rest in complete RPMI.

### 2.5. Caco-2 and THP-1 Co-Culture Model

Caco-2 cells were seeded onto transwell insert plates at a concentration of 5 × 10^4^ cells/well and cultured for 21 days until the cells were fully differentiated. The culture medium was changed every 2–3 days until Caco-2 cells were fully differentiated into polarized monolayers. THP-1 cells were seeded in 12-well plates (2 × 10^5^ cells/well) with PMA (150 ng/mL) for 48 h. After replacing the medium, inserts with Caco-2 cell monolayers were added to the plates containing THP-1 cells. Next, 1 μg/mL LPS (Sigma-Aldrich, St. Louis, MO, USA, cat. no. L5753) was added to the basolateral side, and different concentrations of BWI mix were added to the apical side. After 48 h of incubation, Caco-2 cells and the medium from the insert were collected for the inflammatory gene qRT-PCR, ELISA, and western blotting.

### 2.6. Cell Viability Assay

Caco-2 cells (5 × 10^5^ cells/well) were seeded in 6-well culture plates for 48 h in medium with LPS (1 μg/mL) or different concentrations of BWI mix. After incubation, the cell culture medium was replaced twice with PBS and incubated with 5 ug/mL MTT solution (Invitrogen, Waltham, MA, USA, cat. no. M6494) containing serum-free medium for 1 h. After incubation, insoluble formazan was dissolved in DMSO (Sigma-Aldrich, St. Louis, MO, USA, cat. no. D2650). The sample absorbance was measured at 540 nm using a SpectraMax-iD3 microplate reader (Molecular Devices, San Jose, CA, USA). The percentage of cells was calculated relative to that of the control.

### 2.7. Immunofluorescence Microscopy

Caco-2 cells (5 × 10^5^ cells/well) were cultured and treated for 48 h with the control medium, medium with LPS (1 μg/mL), and medium with LPS (1 μg/mL) and BWI mix (10^7^ CFU/mL). After being washed with PBS, Caco-2 cells were fixed with 4% paraformaldehyde in PBS pH 7.4 for 10 min and permeabilized with PBS containing 0.25% Triton X-100 for 10 min at 25 °C. Next, Caco-2 cells were incubated with 1% BSA in PBS for 30 min at room temperature. The primary antibody occludin (Cell Signaling, Danvers, MA, USA, cat. no. #5506) was diluted 1:20 and incubated for 1 h overnight at 4 °C and a secondary antibody (fluorescein isothiocyanate (FITC)-conjugated goat anti-rabbit IgG antibody (Sigma-Aldrich, St. Louis, MO, USA, cat. no. F9887) was diluted to 1:100 and incubated for 30 min at room temperature. Finally, 300 nM DAPI solution (Invitrogen, Waltham, MA, USA, cat. no. D1306) was added to Caco-2 cells and incubated for 10 min at room temperature for nuclear counterstaining. After mounting with a mounting solution (Invitrogen, Waltham, MA, USA, cat. no. 00-4958-02), images were acquired using an Observer D1 inverted microscope (Carl Zeiss, Jena, Thuringia, Germany) equipped with Automatic Component Recognition module and AxioVision 3.1 software (Carl Zeiss, Jena, Thuringia, Germany) using a 40×/1.0 objective lens.

### 2.8. Quantitative Real-Time (qRT)-PCR

To determine cytokine expression at the mRNA level, total RNA was extracted using a MiniBEST Universal RNA Extraction Kit (Takara, Shiga, Japan, cat. no. #9767). Next, cDNA was synthesized from total RNA (2 µg) using a High-Capacity cDNA Reverse Transcription Kit (Applied Biosystems, Waltham, MA, USA, cat. no. 4368813). qRT-PCR was performed using the SYBR Green PCR Master Mix (Applied Biosystems, Waltham, MA, USA, cat. no. 4309155) on a QuantStudio 6 Flex Real-Time PCR system (Applied Biosystems, Waltham, MA, USA). The oligonucleotide primers used were as follows: TNF-α, forward 5′-ACAAGCCTGTAGCCCATGTT-3′ and reverse 5′-AAAGTAGACCTGCCCAGACT-3′; IL-1β, forward 5′-GGATATGGAGCAACAAGTGG-3′ and reverse 5′-ATGTACCAGTTGGGGAACTG; IL-8, forward 5′-TTGGCAGCCTTCCTGATT-3′ and reverse 5′-AACTTCTCCACAACCCTCTG-3′; β-actin, forward 5′-ATTGCCGACAGGATGCAGAA-3′ and reverse 5′-AAGCATTTGCGGTGGACGAT-3′. The thermocycling conditions were as follows: 95 °C for 10 min and 40 cycles of denaturation at 95 °C for 15 s, followed by annealing and extension at 60 °C for 1 min. Fold changes in gene expression were calculated using the ΔΔCt method and normalized to that of β-actin in each sample.

### 2.9. Enzyme-Linked Immunosorbent Assay (ELISA)

The immunomodulatory efficacy of the BWI mix was determined by measuring IL-8 levels in THP-1 medium from the basolateral side of the Caco-2/THP-1 co-culture model. The IL-8 secretion by inflammatory response was measured using a human IL-8 DuoSet ELISA kit (Bio-techne, Minneapolis, MN, USA, cat. no. DY208-05), according to the manufacturer’s protocols. Optical density was determined at 450 nm photometrically using a SpectraMax-iD3 microplate reader. The concentration of released IL-8 was quantified using relevant standard curves in triplicate.

### 2.10. Immunoblotting

The quantity of anti-inflammatory and tight junction proteins was determined by an automated capillary-based nanoimmunoassay using the JESS^TM^ Simple Western system (ProteinSimple, Bio-Techne, Minneapolis, MN, USA) following the manufacturer’s standard method. Protein extracts (0.5 μg/μL) were mixed with 0.1× sample buffer and fluorescent 5× master mix (ProteinSimple, Bio-Techne, Minneapolis, MN, USA, cat. no. PS-FL01-8) to achieve a final concentration of 0.4 μg/μL in the presence of fluorescent molecular weight markers (ProteinSimple, Bio-Techne, Minneapolis, MN, USA, cat. no. PS-ST01EZ-8). Samples were denatured and equally loaded into a 12–230 kDa Jess/Wes Separation Module (ProteinSimple, Bio-Techne, Minneapolis, MN, USA, cat. no. SM W001). Primary antibodies were used for western blotting at a concentration of 40 μg/mL (1:25 dilution). Primary antibodies used for western blotting included rabbit anti-occludin (Cell Signaling Technology, Inc., Danvers, MA, USA), rabbit anti-Cox2 (Cell Signaling Technology, Danvers, MA, USA, cat. no. #12282), rabbit anti-NFκB p65 (Cell Signaling Technology, Danvers, MA, USA, cat. no. #8242), rabbit anti-pNFκB (Cell Signaling Technology, Danvers, MA, USA, cat. no. #3033), rabbit anti-AMPK (Cell Signaling Technology, Danvers, MA, USA; cat. no. #2532), rabbit anti-pAMPK (Cell Signaling Technology, Danvers, MA, USA; cat. no. #2535), and rabbit anti-GAPDH (Cell Signaling Technology, Danvers, MA, USA, cat. no. #2118). Next, HRP-conjugated anti-rabbit secondary antibody (ProteinSimple, Minneapolis, MN, USA, cat. no. DM-001) in the Jess/Wes Separation Module, and the HRP signal was visualized using peroxyde/luminol-S (ProteinSimple, Minneapolis, MN, USA, cat. no. 043–311). These chemiluminescent digital images in the capillary were evaluated automatically by Compass Simple Western version 4.1.0 software (ProteinSimple, Minneapolis, MN, USA), which calculated the height, area, and signal/noise ratio in triplicate.

### 2.11. Transepithelial Electrical Resistance (TEER) Measurement

To determine the effects of the BWI mix on the integrity in Caco-2 cells, trans-epithelial electrical resistance (TEER) measurements were performed using an EVOM3 (WPI, Sarasota, FL, USA). Caco-2/THP-1 cells were differentiated and treated with LPS or BWI mix, as described in the Caco-2 and THP-1 co-culture Model. TEER measurements were performed immediately after medium replacement with DMEM containing LPS and/or BWI mix. After 24, 48, and 72 h of incubation, the TEER value was calculated as TEER Ω ∙ cm^2^, where TEER Ω cm^2^ is the TEER of an epithelial layer after subtraction of the TEER of the membrane without a cellular layer, R_measured_ (Ω) is the resistance of the membrane with a cellular layer, and R_membrane_ is the resistance of the membrane measured in the absence of a cellular layer.
TEER _layer_ = {R_measured_ − R_membrane_} × membrane area

Moreover, measurements were expressed as a percentage of TEER value at time 0 (100%) for calibration of the Initial TEER value difference from each co-culture plate. All experiments were repeated at least three times in triplicate within individual experiments.

### 2.12. Statistical Analysis

Data were evaluated using GraphPad Prism software (version 5.0; GraphPad Software, San Diego, CA, USA) and presented as box plots of at least three independent experiments. Independent experiments referred to as n represent the number of independent cell culture preparations. The Kolmogorov–Smirnov normality test was performed to test whether the values came from a Gaussian distribution. Statistical comparisons of vehicle controls versus treatment were performed using one-way analysis of variance (ANOVA) with Tukey’s comparisons test as the mean ± SD of three independent experiments. The levels of significance are indicated as * *p* < 0.05, ** *p* < 0.01, and *** *p* < 0.001.

## 3. Results

### 3.1. The Viability of Caco-2 Cells by the Treatment of LPS and BWI Mix

Prior to conducting subsequent experiments, we evaluated the possible toxicity of the BWI mix on Caco-2, an intestinal epithelial cell line. To do so, we treated the cells with varying doses of lipopolysaccharide (LPS) and BWI mix for 24 h, and assessed overall viability using the MTT assay. As depicted in Figure 1, the application of the BWI mix did not elicit any harmful effect on the viability of Caco-2 cells across the range of doses tested, nor did LPS treatment. Thus, we proceeded to investigate the effects of the proposed range of the BWI mix on human intestinal epithelial cells.

### 3.2. The Protective Effect of BWI Mix on TJ Function

As LPS is a component of the cell wall of gram-negative bacteria, it is considered the main endotoxin involved in several human intestinal diseases, such as inflammatory bowel disease and necrotizing enterocolitis [30]. LPS binds to intestinal epithelial cells, particularly to the plasma membrane receptor known as TLR4, and activates receptor-mediated host cell signaling, which leads to an increase in the permeability of the epithelial cells, compromising the integrity of the barrier function [31]. Therefore, our first objective was to investigate whether the BWI mix could protective against LPS-induced compromised barrier function in Caco-2 cells. To assess this, we observed the integrity of the plasma membrane of Caco-2 cells by staining for occludin, a TJ protein [32], in the presence or absence of BWI mix after LPS treatment for 48 h. As shown in Figure 2, the integrity of the plasma membrane between adjacent Caco-2 cells was clearly intact in no-treatment control group, while the overall intensity of occludin was significantly reduced in the LPS-treated cells, indicating impaired barrier function. In some cells, t the impairment was particularly severe (shown as white arrows in Figure 2A) when the cells were exposed to LPS alone, indicating increased permeability due to loosening TJs between the cells. In contrast, treatment of the BWI mix and LPS together resulted in considerable preservation of immunofluorescent staining for occludin, almost comparable to the no-treatment control cells, suggesting that the plasma membrane was protected from LPS damage. To verify the protein levels of occludin and other signaling molecules, the western blots were performed (Figure 2B) under the previous experimental conditions. The data revealed that the protein level of occludin in both LPS and BWI mix treated cells was almost equivalent to the levels of -treated cells. In contrast, the level of occludin was drastically diminished by LPS. Moreover, the level of phosphorylated AMPK, which indirectly measures the assembling TJs, was also reduced by LPS treatment. Interestingly, the amount of phosphorylated AMPK was considerably preserved (~50% compared to untreated cells, Figure 2C) in both LPS and BWI mix treated cells, suggesting BWI mix has a protective role on LPS-induced occludin degradation by interfering with LPS-induced inactivation of AMPK signaling in Caco-2 cells (Figure 2B,C).

### 3.3. The Reduction of LPS-Induced Inflammation by BWI Mix in Co-Culture Model

Previously, we demonstrated that BWI mix is effective in protecting the cellular barrier of epithelial cells from LPS-induced damage. To better understand the role of BWI mix, we utilized an in vitro co-culture model system that mimics the real condition of gut structure.

Caco-2 cells were allowed to spontaneously differentiate into a monolayer of intestinal epithelial cells over a period of 21 days, resulting in the formation of polarized cells with apical and basolateral substructures. THP-1, a human leukemia monocytic cell line, was treated with PMA to differentiate the cells into macrophages to mimic the immune cells in the lamina propria. After differentiation, the Caco-2 cells on the transwell insert were placed onto the top of the differentiated THP-1 cells. BWI mix was added onto the apical side of Caco-2 cells, and LPS was applied to the basolateral side of Caco-2 cells physically separated from THP-1 cells by the transwell insert. However, the interaction between two different cell types was possible through the exchange of soluble factors released from the cells.

By utilizing this in vitro co-culture model system, we investigated whether BWI mix relieves inflammation in intestinal epithelial cells caused by proinflammatory (M1-like) macrophages. To measure the degree of inflammation caused by LPS, we analyzed the relative mRNA expressions for four different proinflammatory cytokines (Cox-2, TNF-α, IL-1β, and IL-8) from the differentiated Caco-2 cells by performing qRT-PCRs. As a result, we identified a general increase in mRNA levels for all proinflammatory cytokines with the LPS challenge as expected (Figure 3A–D) However, the mRNA levels of those LPS-sensitive proinflammatory cytokines s were dose-dependently decreased with two dosages of BWI mix indicating that the inflammation caused by LPS in the epithelial cells was alleviated by the treatment of BWI mix.

This finding was supported by subsequent experiments under the same conditions. First, Nuclear Factor Kappa B (NFκB) activations were monitored as shown in Figure 3E. In this immunoblotting data, BWI mix apparently attenuated the degree of activation (phosphorylated) of NFκB compared to LPS alone. This finding suggests that the observed moderated levels of activation of NFκB are correlated with previous qRT-PCR results, which showed the reduction of LPS-induced inflammation. Finally, we compared the levels of secreted IL-8 proteins from the cells by performing ELISA, and LPS-induced IL-8 secretion was considerably decreased (24.9% reduction to the LPS alone) by the low dosage of BWI mix.

### 3.4. The Measuring TEER to Assess the Integrity of Intestinal Epithelial Barrier Function in Co-Culture System

Previously, we identified the protective effects of BWI mix on the integrity of the plasma membrane by preserving occludin from LPS-induced damage and suggested that the activation of the AMPK pathway by BWI mix is a potential mechanism. However, this evidence is not a measurement of the barrier function but provides indirect confirmation of the integrity of the epithelial barrier function. Therefore, we utilized the TEER assay to provide direct proof for measuring the integrity of TJ dynamics in a Caco-2/THP-1 co-culture model.

TEER measurements were performed in the same co-culture model system used in the previous experiment, but a time course applied. As shown in Figure 4, LPS treatment alone led to a drastic decrease in TEER after 24 h and exhibited the lowest TEER value (55% compared to the control) at 48 h. LPS-induced lowering of TEER values did not recover until 72 h compared to the normal control, implying that the barrier function was seriously damaged, in part, by LPS-induced activation of differentiated THP-1 cells, which mimic M1-like macrophages. In addition, the damage by LPS was attenuated by the addition of BWI mix, and recovery was observed in a time- and dose-dependent manner. Moreover, the recovery rate by the high dose of BWI mix increased rapidly from 48 h to 72 h, and the TEER observed after 72 h was apparently recovered up to 87.2% compared to the normal control. Taken together, these findings strongly support the idea that the BWI mix is effective in protecting intestinal epithelial cells by enhancing the integrity of the barrier function of the cells when they are exposed to the inflammatory environment mediated by invading pathogens.

## 4. Discussion

Microbiome technology has brought considerable alternatives for finding better solutions to improve individual health [33,34,35]. Including numerous basic studies performed using cell lines and animals, human clinical trials with various indications have been conducted to understand the potential benefits of probiotics, and the radius of application is rapidly increasing [36,37]. The consumption of probiotics for better health is not restricted to certain age populations, but is applicable for all age populations. The consumption of probiotics has increased over the last two decades in the US [38]. Although the precise mechanism of action of probiotics is not fully understood, the intestine is the main organ for ingested microbes to excise the benefits to the entire body [39].

In this study, we investigated whether BWI, a commercially available probiotic formular composed of multiple species (eight strains and a postbiotic material), is effective for alleviating inflammation of Caco-2 cells. The testing hypothesis was the probiotic mix potentiate TJ function of intestinal epithelial cells leading to strengthen barrier function when they are challenged by inflammatory agents. The significance of our study is that we have shown the effectiveness by utilizing multi-species formula. The majority of the studies of probiotic strains have focused onto single strains to investigate specific effectiveness in various indications, however commercial probiotic supplements are preferred to be composed of multiple species with some exceptions for practical reasons. Each species (even strain) has its unique characteristics such as the survival rate in the body, the adherence to the epithelial surface, the colonizing place, the antimicrobial activity, and the metabolites. Thus, there have been plentiful efforts to formulate ideal probiotic supplements by mix and match of them expecting multiple functionalities or even synergy effects. However, formulating probiotic supplements is not a straightforward task due to the complex interactions between species (strains), which can result in reduced effectiveness and undesirable side effects [40,41,42]. In addition to functional considerations, there are also non-functional factors to consider in the commercial formulations, such as stabilities, productivities, and governmental regulations. Therefore, it is not necessarily reasonable to expect the same or comparable effects from commercial probiotic formulations that contain functionally proven strains mixed with other strains for other reasons.

Several reliable scientific reports have demonstrated the effectiveness of multi-species probiotic mixtures in promoting intestinal barrier function (references omitted). However, to the best of our knowledge, most of the probiotic products that are manufactured according to the original formulas mentioned in the literature have not been developed to an industrial scale. The commercial multi-species mixture VSL#3, which was invented by Dr. Claudio De Simone, is the only formula claiming to promote intestinal barrier function that has been supported by both in vitro and in vivo results [43,44,45].

In this study, we present experimental evidence for the anti-inflammatory effects of BWI, a multi-species probiotic formula that has been developed into an industrial product, in gut-derived epithelial cells. Throughout the entire experiment, the numbers of BWI mix on Caco-2 cells were validated by the observation of no toxicities after the BWI mix treatments (Figure 1). We investigated whether BWI mix can alleviate LPS-induced inflammation in Caco-2 cells. As shown in Figure 2A, the application of BWI mix to the cells maintained the overall integrity of plasma membranes between adjoining cells even when the cells were inflamed by LPS. This finding was supported by the immunofluorescence imaging, which indicated the preserved amount of occludin, a major protein in TJ complexes. These results suggest that the BWI mix treatment protected the barrier between adjoined cells from LPS-induced damage. Furthermore, the western blot (Figure 2B) showed an almost equivalent amount of occludin compared to the control group (untreated cells) when the cells were treated with both LPS and BWI mix together. In addition, the immune blot confirmed that the level of the activated form of AMPK, which is phosphorylated, was considerably protected by the BWI mix treatment, while LPS-induced AMPK activation was diminished. Since the activation of the AMPK pathway is known to mediate functional TJ assembly [46,47,48,49], we speculated that the maintained amount of occludin in LPS-inflamed cells in the presence of BWI mix was caused by the activation of the AMPK pathway. The results suggest that the BWI mix treatment may have anti-inflammatory effects on gut-derived epithelial cells, and may help intensify the integrity of the cell barrier function.

Since the Caco-2 model does not fully replicate the complexity of the gut, which is composed of different types of cells, we modified the model by using a co-culture system with two differentiated cell lines (Caco-2 and THP-1) to better mimic the structure of the human gut [50,51,52]. Caco-2 cells were grown as a mono-layered intestinal epithelium, while THP-1 cells, which are M1-like macrophages, were used as immune cells to respond to the inflammatory environment of the gut through various stimulations such as LPS, a major cell wall component of gram-negative bacteria that is generally considered harmful to the human gut. Using this in vitro co-culture model, we found that BWI mix treatment relieved the inflammation induced by LPS, as evidenced by the reduced expression of COX-2 and proinflammatory cytokines (Figure 3A–D). Additionally, we observed that the activation (phosphorylation) of NFκB signaling was inhibited in BWI mix treated cells compared to cells treated with LPS alone (Figure 3E). When in an inactive state, NFκB is bound to its inhibitory protein, IκB, and located in the cytoplasm. Upon stimulation by external stimuli, IκB is degraded, leading to the release and translocation of phosphorylated NFκB to the nucleus. Once in the nucleus, NFκB binds to specific DNA sites and initiates the transcription of proinflammatory cytokines [53,54]. These findings suggest that the BWI mix may have potential therapeutic effects in alleviating gut inflammation and improving gut health.

TEER measurements were originally developed as an alternative to tracer-compound-based methods to evaluate drug permeability, since the tracer itself can affect the barrier function [55,56]. By monitoring the electrical resistance, which is dynamically affected by the ionic conductance between the monolayers of cells, a more reliable and reproducible measurement of the integrity of the epithelial barrier function is achievable. The TEER test for BWI mix in the co-culture model indicated that BWI mix improved the TEER, which is a measure of the integrity of the barrier function (Figure 4). In summary, our findings provide additional in vitro evidence for the effectiveness of a commercial multi-species probiotic supplement in improving intestinal barrier function.

Our study utilized an in vitro model system with inherent limitations, such as the in vitro co-culture model system which does not fully capture the complexity of the human intestine. This model system comprises only two types of cells and lacks important components of the tissue microenvironment that may play a role in the inflammatory process. In addition, the routes of administration of probiotics should be carefully considered, as the viability of live probiotic microbes from the mouth through the intestine can greatly influence their effectiveness, given their sensitivity to gastrointestinal enzymes and digestive juices. Thus, caution is necessary when interpreting our study results in the context of real human cases. Nonetheless, our findings can provide a valuable basis for generating hypotheses and informing the development of more relevant in vivo model systems, including human clinical trials.

## 5. Conclusions

Our study provides evidence that BWI mix can enhance the integrity of epithelial barrier function in the presence of LPS-induced inflammation. Specifically, we found that BWI mix can potentiate the signaling pathway involved in assembling TJs complexes, which are essential for maintaining the integrity of the epithelial barrier. In addition, we observed that BWI mix can reduce LPS-induced proinflammatory gene expressions, further supporting its protective effects against inflammation. Our TEER analysis, which utilized an in vitro co-culture model system, demonstrated a dose-dependent protection in the cellular permeability, suggesting that BWI mix can reduce damage to the epithelial cells in an inflammatory environment. Further studies will focus on identifying the precise mechanisms underlying the protective effects of BWI mix, as well as exploring its potential as a probiotic supplement with other potential functionalities.

## Figures and Tables

**Figure 1 microorganisms-11-00656-f001:**
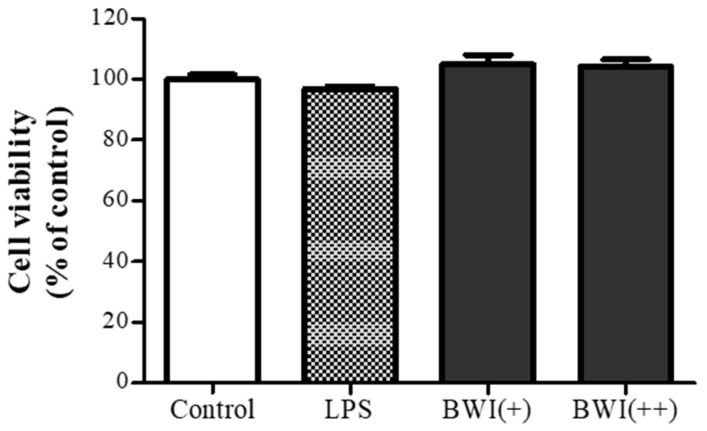
The viability of Caco-2 cells by LPS and BWI mix. The viability of cells was compared by treating them with LPS (1 ug/mL) and two doses of BWI mix (“+” and “++” were 1 × 10^6^ and 1 × 10^7^ cells/mL respectively). The data are presented as the mean ± standard deviation (SD) of three independent experiments.

**Figure 2 microorganisms-11-00656-f002:**
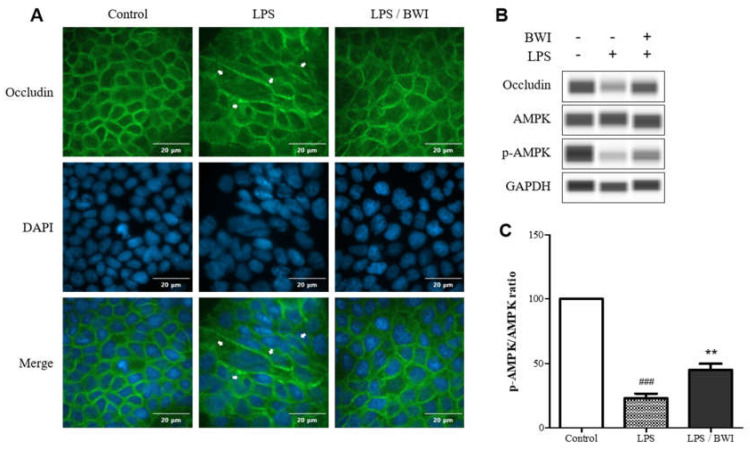
BWI mix mediated protection of occludin on plasma membrane and AMPK activation. (**A**) The top panel shows immunofluorescence images of occludin (green), the middle panel shows DAPI staining for nuclei (blue), and the the bottom panel shows the merged images in the presence or absence of LPS (1 ug/mL) and/or BWI mix (1 × 10^7^ cells/mL) Cellular barrier damages induced by LPS are indicated by white arrows (**B**) A representative western blot analyzing relative levels of occludin, AMPK, phosphorylated (activated) AMPK, and GAPDH (loading control) (**C**) The ratios of activated (*p*-AMPK) versus total AMPK in the immunoblots were quantified using Compass Simple Western software (### *p* < 0.001 vs. control; ** *p* < 0.01 vs. LPS).

**Figure 3 microorganisms-11-00656-f003:**
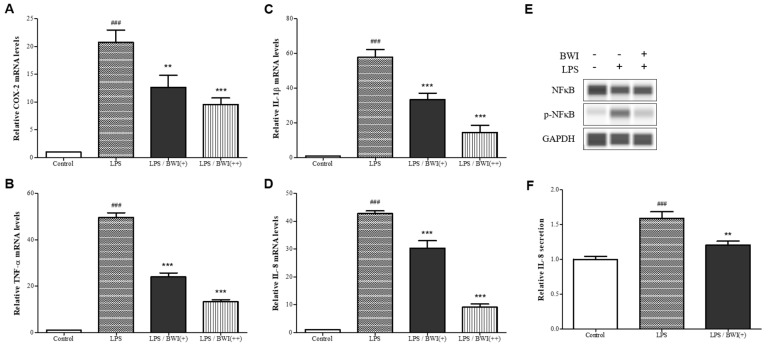
The reduction of LPS-induced inflammation in mono-layered Caco-2 cells co-cultured with differentiated THP-1 cells by BWI mix. (**A**–**F**) The relative mRNA expression levels of Cox-2, TNF-α, IL-β, and IL-8 were analyzed in the presence or absence of LPS and/or two doses of BWI mix (“+” and “++” were 1 × 10^6^ and 1 × 10^7^ cells/mL respectively). (**E**) Representative western blot data showing relative levels of NFκB, phpsphorylated-NFκB, and GAPDH (loading control) in harvested Caco-2 cells. (F) Secreted levels of IL-8 in the media were measured by ELISA. ^###^ *p* < 0.001 vs. control; ** *p* < 0.01 vs. LPS; *** *p* < 0.001 vs. LPS.

**Figure 4 microorganisms-11-00656-f004:**
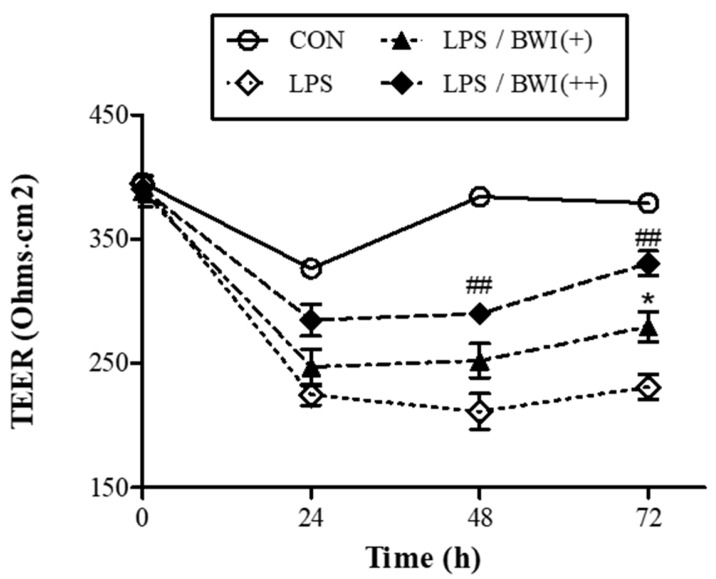
BWI mediated-increase in barrier integrity measured by TEER in the co-culture model system. By measuring TEER, the integrity of intestinal barrier function was observed for a given time course. The co-culture model system applied in previous experiments was used to monitor the TEER value. The LPS (1 ug/mL) and two doses of BWI mix (“+” and “++” were 1 × 10^6^ and 1 × 10^7^ cells/mL respectively) were applied to the experiments. ^##^ *p* < 0.01 vs. control; * *p* < 0.05 vs. LPS.

## Data Availability

All relevant data are provided in the form of regular figures and tables.

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
