# Peer review of "Multi-Species Probiotic Strain Mixture Enhances Intestinal Barrier Function by Regulating Inflammation and Tight Junctions in Lipopolysaccharides Stimulated Caco-2 Cells"

_microorganisms, 2023, doi:10.3390/microorganisms11030656_

Round 1
Reviewer 1 Report
Minor points:
References 3 and 4 there is a blank space.
It seems to me that there is some redundancy in “In general, consuming probiotic microbes as,…..”
In the preparation of the probiotic mixture, what is the total amount of CFU in the BWI formula?.
In the discussion: wouldn't it be better to use large intestine o large bowel instead of gut intestine?
Is there a validation plan for all equipment used that ensures that the results are reproducible?
Major point:
All researchers in this work are employees of Lactomason Co., Ltd. In addition, funding has also been provided by Lactomason Co., Ltd.
So, could the credibility of these results be questioned?
Would it not have been more appropriate for the Lactomason company to give its raw material (probiotics mixture) to an external research group so that the results could be better appreciated by the scientific community?
Conflicts of interest raise reasonable doubts about the interesting conclusions of this work.

Author Response
Dear Reviewer 1,
Please see the attachment.
I appreciate your effort to review our work.
Thank you.

Reviewer 2 Report
Dear Authors,
The concept of this paper was really interesting since the effect of this multistrain probiotic formulation, which is easily accessible by the public and not a random laboratory synthesized microbiological combination, has not been previously described on in vitro models of human pathological conditions. Furthermore, the need for this study is well established based on the bibliography cited, and the scientific protocols used adequately provide answers to the questions asked by the researchers, and follow a thorough, well-organized and well-executed study design.
Having said that, however, there is still room for improvement:
1) Since the probiotic mixture already contained live Lactiplantibacillus plantarum bacteria, why was further heat killed material of this strain included in the experiments? Is this how the formulation is provided by the manufacturer, or was it added by the researchers?
2) Besides the already mentioned fact that it is a mixture and is preferable to single strains, the reason why these specific probiotic strains were used must be included, as well as the concentrations of each of them in the mixture.
3) The Discussion paragraphs seem a little abrupt and would benefit from a wider analysis of the main reasons of the conduction of the study, as well as its novelty and scientific importance of its findings among similar papers, instead of just citing more information about the reasoning behind the use of probiotics in general.
4) Also, some small changes include the sentence “The results are shown in Fig. 2. By treating for details on figures and tables: Caco-2 cells with the BWI mix, which were inflamed by LPS-induced inflammation, the overall integrity of plasma membranes between adjoining cells was improved compared to the LPS-treated cells.” in the second paragraph of the Discussion, which must be better rephrased.
5) Last but not least, the figure legends should become clearer, since they are not currently easily discerned from the main text of the results. In addition, there is too much discussion-related content in the results and methods-related content figure legends, which should both have a more descriptive rather than explanatory form. As I mentioned above, the cited information reported in the Results section should be moved to the Discussion and be analyzed there in even more detail.
Author Response
Dear Reviewer2,
Please see the attachment.
I appreciate your effort for reviewing our work.
Thank you.

Round 2
Reviewer 1 Report
Dear authors:
With the corrections made and yours answers I am satisfied with the conclusions of your interesting article.